# Face mask use in the city of Chennai, India: Results from three serial cross-sectional surveys, 2021

Murali Sharan[1]◉*, Manikandanesan Sakthivel[2]◉, Polani Rubeshkumar[1], Ramya Nagarajan[1], Vettrichelvan Venkatasamy[1], Sendhilkumar Muthappan[1], Mohankumar Raju[2], Joshua Chadwick[1], Kalyani S.[2], Navaneeth S. Krishna[1], Mogan Kaviprawin[2], Pavan Kumar Gollapalli[2], Srinath Ramamurthy[2], Parasuraman Ganeshkumar[1], Murugesan Jagadeesan[3], Prabhdeep Kaur[1]

1 Division of Noncommunicable Diseases, ICMR National Institute of Epidemiology, Chennai, Tamil Nadu, India, 2 SafetyNet, Mumbai, India, 3 Greater Chennai Corporation, Chennai, Tamil Nadu, India

◉ These authors contributed equally to this work.
* sharanmurali@gmail.com

## Abstract

### Introduction

The Government of Tamil Nadu, India, mandated wearing face masks in public places to combat the COVID-19 pandemic. We established face mask surveillance and estimated the prevalence of appropriate mask use (covering the nose, mouth, and chin) in the slums and non-slums of Chennai at different time points in 2021.

### Methods

We conducted three serial cross-sectional surveys in the outdoors and indoors of Chennai in March, July, and October 2021. We observed the mask wearing among 3200 individuals in the outdoors and 1280 in the indoors. We divided the outdoor and indoor locations into slums and non-slums. In October 2021, we also surveyed 150 individuals from each of the 11 shopping malls in the city. We calculated the proportions and 95% confidence interval (95%CI) for the appropriate mask use in the outdoor, indoor, and malls by age, gender, region, and setting (slum and non-slum).

### Results

We observed 3200 individuals in the outdoor and 1280 individuals in the indoor setting, each from a slum and non-slum, during the three rounds of the study. In outdoor and indoors, males comprised three-fourths and middle-aged individuals were half the study population. Mask compliance changed significantly with time (p-value <0.001). Males consistently demonstrated better compliance in all rounds. The south region had the highest mask compliance in slums indoors and outdoors in rounds 4 and 5. Young adults had the highest mask compliance in both outdoor slums and non-slums in all rounds. Overall mask compliance in shopping malls was 57% (95% CI: 48–65).

https://data.mendeley.com/datasets/vkv4cwhh7n
(DOI: 10.17632/vkv4cwhh7n.1).

**Funding:** This study was supported by the
Intramural fund of ICMR-National Institute of
Epidemiology, Chennai, India.

**Competing interests:** The authors have declared
that no competing interests exist.

## Conclusion

The mask compliance in Chennai outdoors during the COVID-19 pandemic was less than 50%, with variations across time points by gender, age groups, and geographical locations. We must develop more effective communication strategies for older age groups and crowded indoor settings.

## Introduction

The SARS-CoV-2 virus transmits from an infected individual to a susceptible individual via droplet, aerosol, or fomite transmission [1,2]. However, the risk for COVID-19 transmission may depend on population density, ventilation, face mask usage, total contact duration, and type of activity (talking, singing, etc.) [3]. Face mask usage showed a high protection rate of 70% against SARS-CoV-2 infection [4]. Considering this available evidence, public health agencies, including World Health Organisation (WHO), the U.S. Centers for Disease Control and Prevention (CDC), and various governments across the world, advocated the use of face masks in all public settings as the prime mitigation strategy [5–7]. High-Income Countries used compliance to face mask use as an ecological indicator in understanding the waxing and waning of the COVID-19 waves [8].

Chennai is the capital city of Tamil Nadu, a southern state of India. Chennai is also the fourth largest metropolitan city in the country, spread over an area of 426 sq. km and a population of nearly 8 million (2020). During the COVID-19 pandemic, Chennai recorded 5,66,147 cases and 8,588 deaths till December 25, 2021 [9]. In alignment with the guidelines issued by the Ministry of Health & Family Welfare (MoHFW), the Government of Tamil Nadu issued mandates for wearing face masks in public places. Violators could be charged a fine of Rs. 500 [10,11].

In 2020, the Greater Chennai Corporation (GCC) and ICMR NIE established the mask compliance surveillance system in Chennai. The team conducted two survey rounds in 2020 and documented the mask compliance in the outdoor locations, indoor places, and malls of Chennai City. GCC used the report for public health advocacy through mass media and law enforcement authorities [12]. We continued our periodic surveillance to understand the change in the behaviour of Chennai City's citizens with changing waves of the COVID-19 pandemic. We estimated the compliance with appropriate mask usage in the outdoor and indoor locations of slums and non-slums of Chennai, Tamil Nadu, India, at three different time points in 2021, and compare the trend with the two rounds in 2020. During the last survey in 2021, we documented the mask compliance in the city's shopping malls, in addition to outdoor location and indoor spaces.

## Methods

### Design & setting

We conducted serial cross-sectional surveys among the residents of Chennai to quantify compliance to face mask usage. We previously completed two rounds in 2020 and published elsewhere [12]. We conducted round 3 at the start of the second COVID-19 wave from March 20 to 26, 2021 to document the mask compliance in rise of a pandemic wave. We conducted round 4 from July 8 to 10, 2021 at the decline of the second COVID-19 to see if the mask wearing pattern has changed. We conducted the last (fifth) round towards the end of the year 2021,

from October 30 to November 1, 2021 to document how the residents adhered to the mask-wearing guidelines when the number of COVID-19 cases were low.

Chennai is a cosmopolitan city divided into 15 zones and 200 wards. The city has around 45,000 streets stratified as Slum and Non-slum areas, mapped and documented for administrative purposes by the Greater Chennai Corporation (GCC). The population density was higher in the slums than in the non-slums. From time to time, the state's public health department released guidelines about compliance with face mask use in public places [10,13].

## Sample size and sampling

Round 1 of the Chennai mask study reported compliance to face mask use in outdoor locations as 36% [12]. Considering this, we calculated the sample size for the three-time points in this study with a 95% confidence interval, 5% absolute precision, and a design effect of 4.5 as 1600. Considering that we will observe 50 individuals from each street, we needed 32 streets each from slums and non-slums from among the 45,000 streets in Chennai. We selected the 32 streets by simple random sampling using Microsoft Excel [14].

We calculated the sample size separately for the indoor locations. Round 2 of the Chennai mask study documented indoor mask compliance as 11% [12]. Using this prevalence, we calculated the sample size with 95% confidence, 5% absolute precision, and a design effect of 4.25 as 640. We chose the indoor locations from the 64 outdoor streets already chosen for the outdoor survey. Thus, we observed 20 individuals in the indoor locations from each of these 64 locations for the indoor survey.

Round 2 of the Chennai mask study reported 57% compliance to face masks in the Chennai malls [12]. Using this proportion, we calculated the sample size with a 95% confidence interval, 5% absolute precision, and a design effect of 4.5 in Open Epi version 3.1 to be 1695 [15]. However, considering that this is a follow-up of round 2 and due to feasibility, we decided to sample 150 individuals from each mall, summing to 1650 individuals in total. We selected the locations using a simple random technique using random number allocation in Microsoft Excel [14].

## Operational definitions

"Mask" was defined as any cloth mask, medical mask, or N95 respirator worn over the face. We defined appropriate mask use as wearing a mask that covers the nose, mouth, and chin [12]. For ease of observation and recording, we grouped the individuals into four age groups: Children or adolescents, young adults, middle age, and old age. We defined streets in residential or commercial areas and bus stations as outdoor public places. We defined closed public settings open to the people as indoor public settings. Most of the indoor public settings in Chennai lacked personnel who would insist on the use of face masks in their shop/establishment. We included all such locations (e.g., grocery shops, vegetable shops, pharmacies, religious places, and apparel stores) as indoor public places. We excluded areas with security personnel who insisted on wearing a mask before entry, and food courts, eateries, liquor shops, and tea/coffee cafes from our observation. Shopping malls, however, had personnel who would insist on mask compliance and hence were included as a separate setting.

## Study procedure

We trained five teams of two data collectors in the operational definitions and conducted simulation exercises to observe face mask compliance directly [8,9]. We aimed to reduce the inter-observer variation through this training and simulation exercise. The technical team mapped the selected locations using Google Maps and shared it with the data collection teams. The

geographical spread of the three survey rounds is shown in Fig 1. Upon reaching the specific tagged location, the stationary data collector would observe the individuals crossing from right to left direction, one individual at a time, and record the individual's mask compliance. We maintained this directional rule to minimize possible duplication. We included pedestrians, bicycle riders, motorcycle riders, autorickshaws, and bus passengers, excluding individuals traveling in a car, and riders wearing a helmet. We repeated this process until the sample size for that location was completed. Individuals in the defined indoor areas and malls were also sampled similarly.

We observed the participants from a distance to avoid the Hawthorne effect. Data collectors recorded age group, gender, and compliance to mask usage as worn appropriately, inappropriately, or with no mask. We recorded the data in an Open Data Kit (ODK), an Android-based mobile application with pictorial cues to reduce the survey time. Observers collected data using their mobile phones, and we received the data in our secured Institutional server. We used the same sampling list and methodology for all three rounds and in malls (round 5 alone) reported in this paper.

### Data analysis

We explored four study settings in the present study–outdoor slums, indoor slums, outdoor non-slums, and indoor non-slums. We summarised the exposure variables age group, gender, and geographical region in Chennai as the frequency with proportions. We summarised the appropriate mask compliance in each of the four settings as proportion and 95% confidence interval (CI). We also computed the mask compliance for each exposure group separately for slum and non-slum in each round. We used the chi-square test to compare mask compliance across the different age groups during each round for the four study settings. We performed a similar analysis for gender and geographical regions in Chennai.

We combined the results of the previously published rounds 1 and 2 into our dataset [17] to compare mask compliance across the five rounds within each setting using chi-square for trend.

We estimated the mask compliance in malls of Chennai during round 5 of the present study. We summarised the mask compliance in each exposure category with its 95% CI. We also used the chi-square test to compare the mask compliance within each exposure category. We performed this analysis separately for slums and non-slums.

We generated the required visualizations to compare the mask compliance across the study groups in the three rounds. We analyzed the data using Survey set analysis in STATA SE (version 17.0) software (StataCorp LLC, College Station, TX, USA) for the study [18]. A p-value less than 0.05 was considered significant.

## Results

We observed 3200 individuals in the outdoor and 1280 individuals in the indoor setting, each from a slum and non-slum, during the three rounds of the study. In indoor and outdoor settings, males constituted three-fourths of the study population. Nearly half of the study population was middle-aged individuals in indoor and outdoor settings. Over one-third (36%) of the indoor and outdoor study population in each round belonged to the northern region of Chennai, followed by central (33%) and south (31%).

### Trends in mask compliance at various stages of the pandemic

Round 4 reported the highest mask compliance of all rounds (non-slum outdoor: 47% [95% CI: 43–53] vs. slum outdoor 41% [95% CI: 34–47], non-slum indoor 33% [95% CI: 27–40] vs.

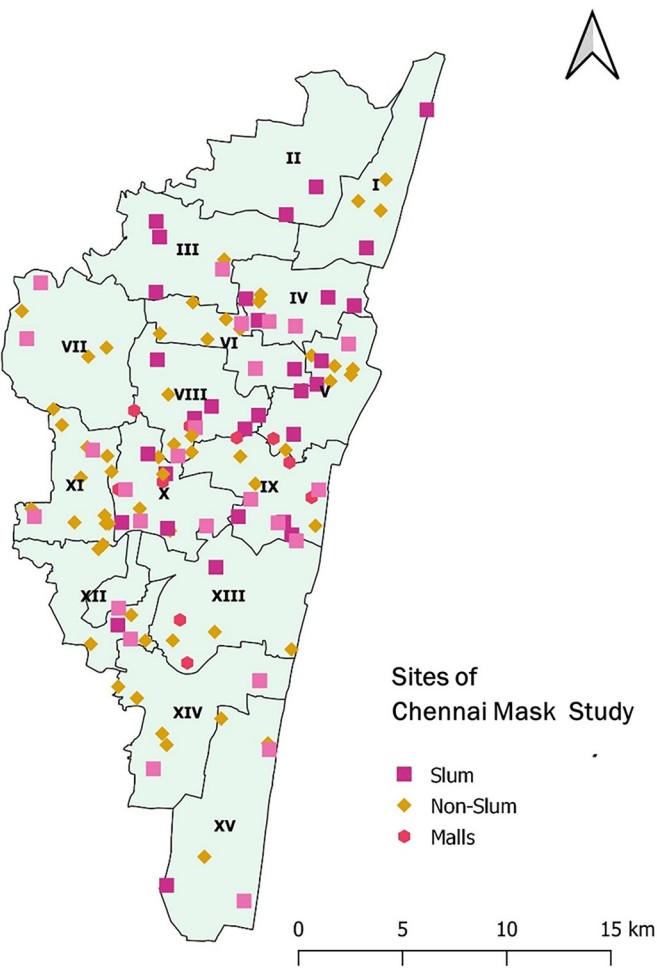

**Fig 1. Study sites of mask survey, Chennai, Tamil Nadu, India, 2021.** We obtained the spatial data to construct the map from the following open-access sources: https://data.opencity.in/dataset/gcc-ward-information [16].

slum indoor 24% [95% CI: 18–31]). Mask compliance was the lowest at round 3, with the compliance in slums indoors reaching the minimum of all times at 11% [95% CI: 8–16] (Fig 2).

The increase or decrease in mask compliance at various survey rounds was not due to chance. There was a significant change in mask compliance with time (p-value <0.001) (Table 1).

## Mask compliance & gender

Mask compliance in the male ranged from 11% (indoor slum–round 3 [95% CI: 8–17]) to 48% (outdoor non-slum–round 4 [95% CI: 43–53]). Whereas among females, mask compliance ranged from the lowest 11% (indoor slum–round 3 [95% CI: 6–19]) to 47% (outdoor non-slum–round 4 [95% CI: 40–56]) (S1 and S2 Tables in S1 File). Males consistently showed higher mask compliance in rounds 3 and 4 in all groups (Fig 3). However, in round 5, females demonstrated better compliance in all settings except outdoor slums, where compliance was lower than men (29% versus 33%). (S1 Table in S1 File).

## Mask compliance & geographical regions

The south region consistently showed the highest mask compliance in slums indoors and outdoors in surveys 4 and 5. In outdoor non-slums, we observed a mixed pattern. In round 3, the

## Compliance to appropriate usage of face mask at various time points during the COVID-19 pandemic in the city of Chennai, India

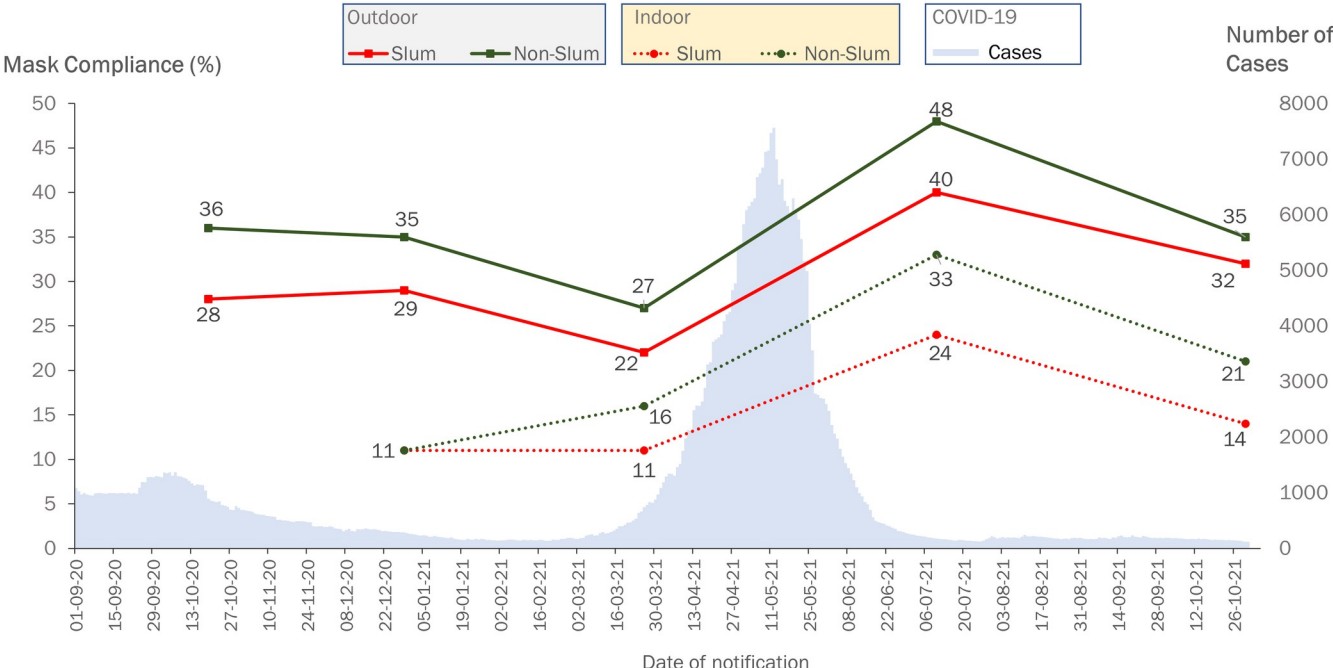

**Fig 2. Compliance with appropriate usage of face masks at various time points during the COVID-19 pandemic in the city of Chennai, India.**

south showed the highest, while in round 4, it was north, and in round 5, it was central (Fig 4). However, in all three rounds, the central region consistently showed higher mask compliance in the indoor non-slums (Fig 4).

**Table 1. Compliance with appropriate face mask usage within each study setting (outdoor/indoor & slum/non-slum) in Chennai compared across the five rounds.**

### OUTDOOR

| Round | Time Period | Slum | | | | | Non-Slum | | | | |
|---|---|---|---|---|---|---|---|---|---|---|---|
| | | N | n | (%) | 95% CI | p-value for trend# | N | n | (%) | 95% CI | p-value for trend# |
| 1 | Oct 2020* | 1800 | 497 | (28) | 23–33 | <0.001 | 1800 | 643 | (36) | 31–41 | 0.002 |
| 2 | Dec 2020* | 1600 | 460 | (29) | 24–35 | | 1600 | 561 | (35) | 31–39 | |
| 3 | Mar 2021 | 1600 | 344 | (22) | 17–27 | | 1600 | 435 | (27) | 23–33 | |
| 4 | Jul 2021 | 1600 | 646 | (41) | 34–47 | | 1600 | 767 | (47) | 43–53 | |
| 5 | Oct–Nov 2021 | 1600 | 518 | (32) | 28–37 | | 1600 | 563 | (35) | 29–41 | |

### INDOOR

| Round | Time Period | Slum | | | | | Non-Slum | | | | |
|---|---|---|---|---|---|---|---|---|---|---|---|
| | | N | n | (%) | 95% CI | p-value for trend# | N | n | (%) | 95% CI | p-value for trend# |
| 2 | Dec 2020* | 640 | 72 | (11) | 8–15 | <0.001 | 640 | 67 | (10) | 8–16 | <0.001 |
| 3 | Mar 2021 | 640 | 72 | (11) | 8–16 | | 640 | 104 | (16) | 13–21 | |
| 4 | Jul 2021 | 640 | 153 | (24) | 18–31 | | 640 | 212 | (33) | 27–40 | |
| 5 | Oct–Nov 2021 | 640 | 92 | (14) | 9–22 | | 640 | 135 | (21) | 16–27 | |

*Results from rounds 1 & 2 of the Chennai Mask Study (12).

#Chi-square for trend.

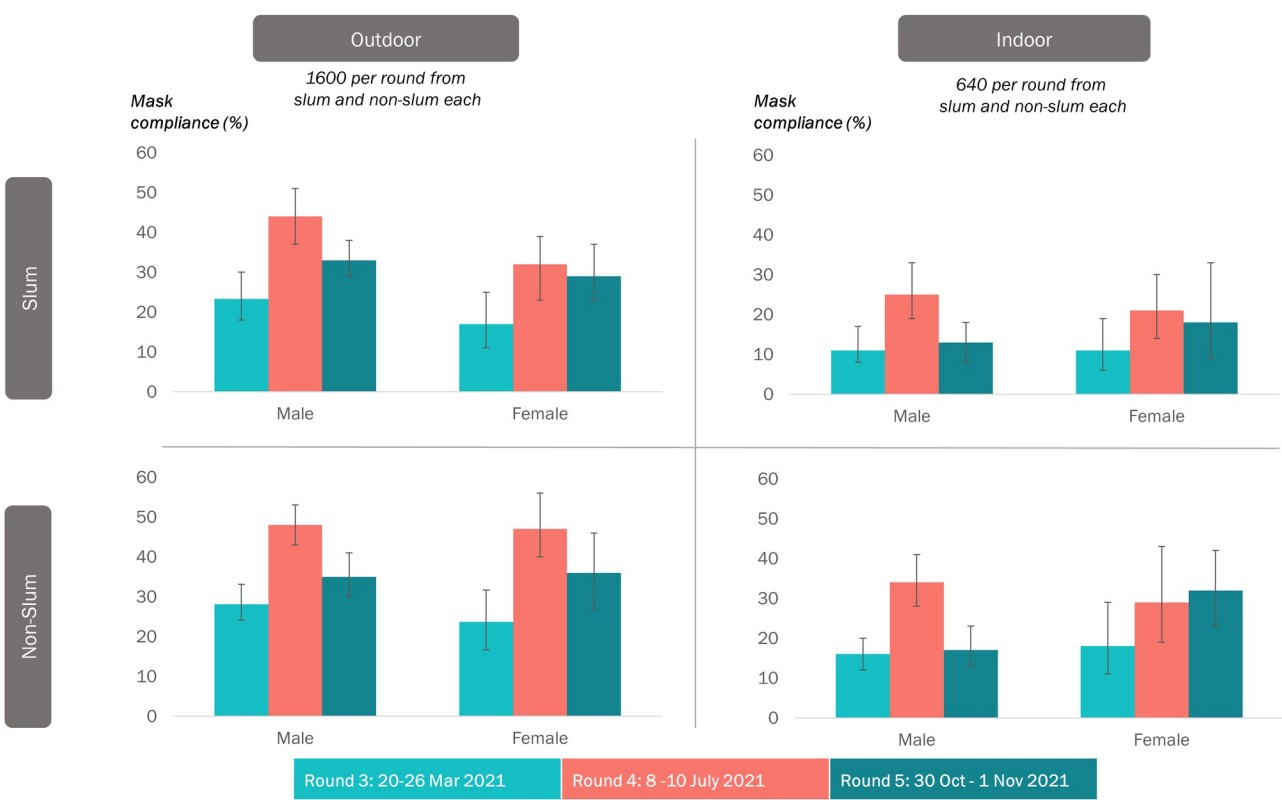

**Fig 3. Compliance with appropriate usage of face masks in the various study settings by gender over the three rounds, Chennai, India.**

### Mask compliance & age groups

Young adults showed the highest mask compliance in both outdoor slums and non-slums. Middle-aged individuals closely followed this. Children/ adolescents showed the lowest mask compliance in round 3 in both slum and non-slum outdoors, which slowly improved over time. In the indoor setting, children demonstrated poor compliance in round 3, which increased consistently to the highest in round 5 (Fig 5).

### Mask compliance in shopping malls (Round 5)

Overall mask compliance in the shopping malls was 57% (95% CI: 48–65) (S1 File). Women (60% [95% CI: 53–67]) showed higher compliance than men (54% [95% CI: 44–64]) in the malls (p-value: 0.061). Older age group individuals had the highest compliance (63% [95% CI: 47–77], followed by children (60% [95% CI: 43–76]). The difference in compliance among the age groups was statistically significant (p-value: 0.039).

### Discussion

The study summarises three repeat surveys to assess compliance with appropriate mask use in Chennai over 2020 and 2021. We also compared the results with the previous two surveys in the same setting using the same methodology [12]. Although the law mandated using face masks in outdoor and closed indoor places, mask compliance remained below 50% outdoors and 33% indoors across the surveys. The present study also documented the low mask compliance in the city's shopping malls during the last survey round. We demonstrated the changes in mask-wearing behavior with time by geographical region, gender, and age group.

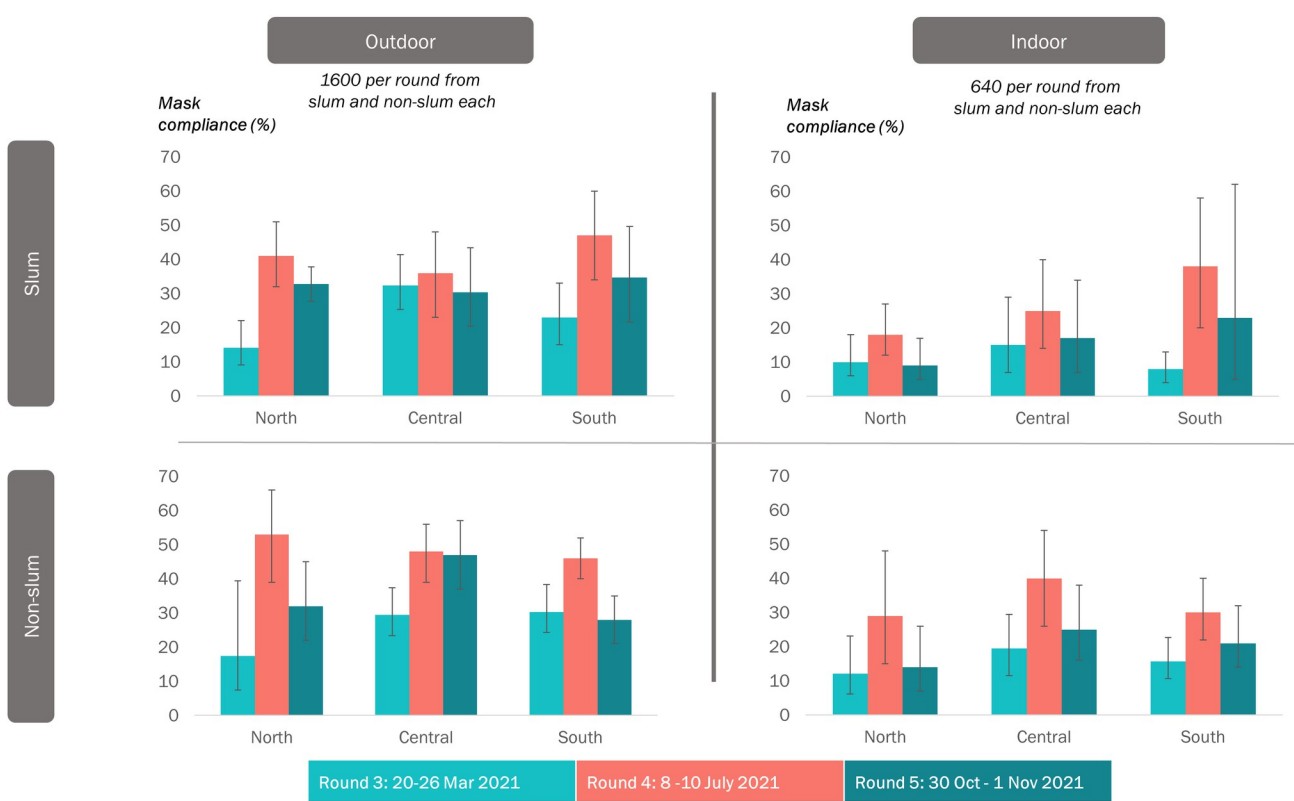

**Fig 4. Compliance with appropriate usage of face masks in the various study settings by geographical regions over the three rounds, Chennai, India.**

The study documented the changes in mask-wearing patterns of the community with changes in overall reported cases of COVID-19. The Government introduced the COVID-19 vaccines in early 2021 [19]. The introduction of the COVID-19 vaccine coincided with the start of the second wave, predominantly driven by the delta variant of SARS-CoV2 [20]. With newer strains in circulation and the poor uptake of COVID-19 vaccines initially, mask use remained the primary mitigation strategy against COVID-19. We documented the highest-ever mask use in the Chennai population during the second wave (July 2021). The present study showed increased mask compliance with increased cases and vice versa. This increase in compliance could be due to the frequent mention of mask use and the increase in cases and deaths frequently mentioned in social and mass media. Evidence from China suggests that mass media could influence health behaviours [21]. In the United States, during the early phase of the Pandemic, the release of CDC recommendations to wear masks led to a 12% improvement in the overall reported mask usage [22]. Although the rules and enforcement were in place in our setting, mask use never exceeded 50% and declined as the number of cases decreased. The only exception was shopping malls where compliance was higher, possibly due to consistent monitoring for masks at the shopping mall [23].

We observed that mask compliance in the outdoor locations was higher than the indoors in all repeat surveys. Literature deems outdoor open spaces as low risk, while indoor closed spaces could be potential high-risk areas for transmission of SARS-Cov2 [24]. This swapping of the intended mask compliance in the population could be due to social appearance anxiety [25]. Law enforcement constantly monitored outdoor locations, which could have led to

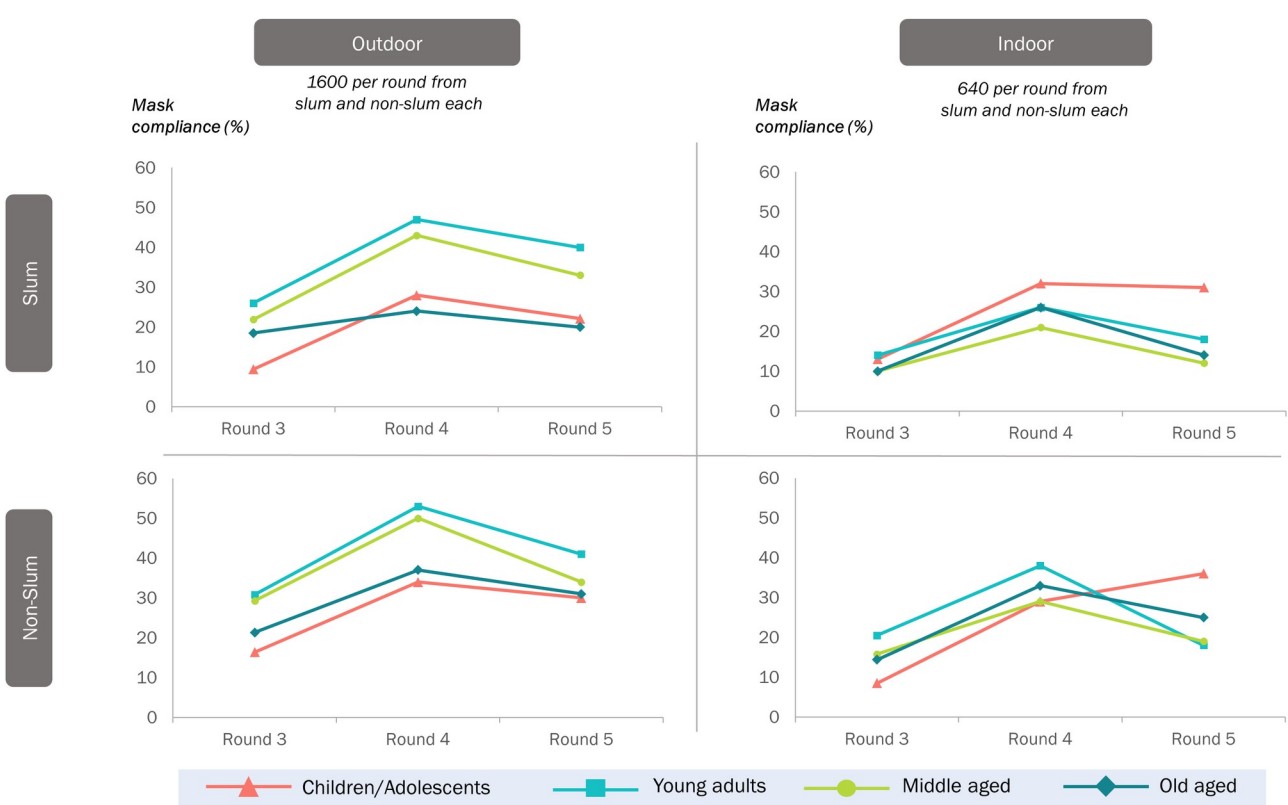

**Fig 5. Compliance with appropriate usage of face masks in the various study settings by age groups over the three rounds, Chennai, India.**

higher compliance. The health promotion messages need to emphasize the high risk of transmission in indoor settings during respiratory illness outbreaks.

We observed that compliance with appropriate mask use was higher among the males than the females in the outdoor locations in most rounds. On the contrary, females had higher mask use in indoor settings. Our results were consistent with a survey from the US, where women entering retail stores during the Pandemic reported higher mask compliance than men [26–28].

In the present study, young adults showed higher mask compliance than other age groups in outdoor settings. A modeling study combined the results of four separate datasets from surveys conducted in high-income settings among MTURK employees, indicating that the older age group is less likely to wear face masks than the young, which also agrees with our study setting [29]. A study from two Swiss hospitals documented the perception of their employees for wearing masks. The young wore masks due to the perceived risk and also for social preferences, while the old wore masks due to only the perceived risk [30]. Studies have proved a positive association between higher self-related risk perception and better face mask compliance [31].

The present study documented higher compliance to appropriate mask use at all time points in the non-slum settlements than in the slum settlements. The non-slum population might have perceived the risk of COVID-19 better hence the consistently high compliance with face masks [32,33]. Also, the lack of social acceptance in non-slum settings could have increased compliance [31]. The southern region of Chennai, which is the hub for information technology companies, showed higher compliance when compared to that of the central and

northern areas [32]. The level of education and social acceptance might play an essential role in mask-use behaviours.

Our study was a rapid and low-cost methodology to monitor mask compliance in public places by direct in-person observation which is one of the recommended methods for measuring compliance to safe behaviours. Our study was able to give mask compliance in different environments in Chennai city–Outdoors, Indoors, and also in Shopping Malls. Our study had some limitations too. In the present study, we anticipate possible inter-observer variation in identifying the individual's age group. But through vigorous training, simulation exercises, and observation rules, we have tried to keep this bias to the minimum.

## Conclusion & recommendation

The mask compliance in Chennai outdoors during the COVID-19 pandemic was less than 50%, with variations across time points by gender, age groups, and geographical locations. Although the COVID-19 pandemic has subsided, the survey findings provide essential insights into mask-use behavior. They are relevant to plan interventions during the resurgence of COVID-19 or other respiratory disease epidemics. We must set realistic expectations about adopting mask use and develop more effective communication strategies. Communication should focus on wearing the mask in crowded indoor settings, and monitoring should be improved in indoor environments. The intensified communication campaigns should focus on older age groups at higher risk and slum areas.

## Supporting information

**S1 File. Tables of face mask compliance in different settings (outdoor, indoor, and shopping malls) and exposure categories (age group, gender, and region) in Chennai, India.**
(PDF)

## Acknowledgments

We thank the Greater Chennai Corporation health staff for providing the necessary support for this project. We acknowledge the contribution of Ramya Kumaraguruparan, Lavanya Ayyasamy, Dharsikaa T, Vijayaprabha Radhakrishnan, Murali Mohan Muni krishnaiah, Suresh Arunachalam, Punita Muni krishna Gandhi, Prakash Marappan, and Ezhil Pounraj for their valuable contributions during the data collection of this study. The authors are grateful for the excellent support provided by Alby John Varghese, Manish Narnaware, and Gagandeep Singh Bedi for this study.

## Author Contributions

**Conceptualization:** Murali Sharan, Manikandanesan Sakthivel, Polani Rubeshkumar, Ramya Nagarajan, Mohankumar Raju, Parasuraman Ganeshkumar, Murugesan Jagadeesan, Prabhdeep Kaur.

**Data curation:** Murali Sharan, Manikandanesan Sakthivel, Polani Rubeshkumar, Ramya Nagarajan, Vettrichelvan Venkatasamy, Sendhilkumar Muthappan, Mohankumar Raju, Joshua Chadwick, Kalyani S., Navaneeth S. Krishna, Mogan Kaviprawin, Pavan Kumar Gollapalli, Srinath Ramamurthy, Parasuraman Ganeshkumar, Murugesan Jagadeesan, Prabhdeep Kaur.

**Formal analysis:** Murali Sharan, Manikandanesan Sakthivel.

**Investigation:** Murali Sharan, Manikandanesan Sakthivel, Polani Rubeshkumar, Ramya Nagarajan, Vettrichelvan Venkatasamy, Sendhilkumar Muthappan, Mohankumar Raju, Joshua Chadwick, Kalyani S., Navaneeth S. Krishna, Mogan Kaviprawin, Pavan Kumar Gollapalli, Srinath Ramamurthy.

**Methodology:** Murali Sharan, Manikandanesan Sakthivel, Polani Rubeshkumar, Ramya Nagarajan, Vettrichelvan Venkatasamy, Sendhilkumar Muthappan, Mohankumar Raju, Joshua Chadwick, Kalyani S., Navaneeth S. Krishna, Mogan Kaviprawin, Pavan Kumar Gollapalli, Srinath Ramamurthy, Prabhdeep Kaur.

**Project administration:** Murali Sharan, Manikandanesan Sakthivel, Polani Rubeshkumar, Mohankumar Raju, Parasuraman Ganeshkumar, Murugesan Jagadeesan, Prabhdeep Kaur.

**Resources:** Vettrichelvan Venkatasamy, Parasuraman Ganeshkumar, Murugesan Jagadeesan.

**Software:** Vettrichelvan Venkatasamy.

**Supervision:** Murali Sharan, Manikandanesan Sakthivel, Parasuraman Ganeshkumar, Murugesan Jagadeesan, Prabhdeep Kaur.

**Validation:** Murali Sharan, Manikandanesan Sakthivel, Polani Rubeshkumar, Ramya Nagarajan, Mohankumar Raju, Parasuraman Ganeshkumar, Murugesan Jagadeesan, Prabhdeep Kaur.

**Visualization:** Murali Sharan, Manikandanesan Sakthivel, Vettrichelvan Venkatasamy.

**Writing – original draft:** Murali Sharan, Manikandanesan Sakthivel, Prabhdeep Kaur.

**Writing – review & editing:** Murali Sharan, Manikandanesan Sakthivel, Polani Rubeshkumar, Ramya Nagarajan, Vettrichelvan Venkatasamy, Sendhilkumar Muthappan, Mohankumar Raju, Joshua Chadwick, Kalyani S., Navaneeth S. Krishna, Mogan Kaviprawin, Pavan Kumar Gollapalli, Srinath Ramamurthy, Parasuraman Ganeshkumar, Murugesan Jagadeesan, Prabhdeep Kaur.

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
