## [Decision Letter · Decision Letter 0]

29 Sep 2023

PONE-D-23-17287Face mask use in the city of Chennai, India: Results from three serial cross-sectional surveys, 2021PLOS ONE

Dear Dr. Sharan,

Thank you for submitting your manuscript to PLOS ONE. After careful consideration, we feel that it has merit but does not fully meet PLOS ONE’s publication criteria as it currently stands. Therefore, we invite you to submit a revised version of the manuscript that addresses the points raised during the review process. Please submit your revised manuscript by Nov 13 2023 11:59PM. If you will need more time than this to complete your revisions, please reply to this message or contact the journal office at plosone@plos.org. Please include the following items when submitting your revised manuscript:A rebuttal letter that responds to each point raised by the academic editor and reviewer(s). You should upload this letter as a separate file labeled 'Response to Reviewers'.A marked-up copy of your manuscript that highlights changes made to the original version. You should upload this as a separate file labeled 'Revised Manuscript with Track Changes'.An unmarked version of your revised paper without tracked changes. You should upload this as a separate file labeled 'Manuscript'.

We look forward to receiving your revised manuscript.

Kind regards,

Vijayaprasad Gopichandran

Academic Editor

PLOS ONE

Journal Requirements:

"This study has not received any specific funding support. The work was supported by the Intramural fund of ICMR-National Institute of Epidemiology, Chennai, India. The funders had no role in study design, data collection, and analysis, decision to publish, or preparation of the manuscript."

4. We note that Figure 1 in your submission contain map images which may be copyrighted. All PLOS content is published under the Creative Commons Attribution License (CC BY 4.0), which means that the manuscript, images, and Supporting Information files will be freely available online, and any third party is permitted to access, download, copy, distribute, and use these materials in any way, even commercially, with proper attribution. For these reasons, we cannot publish previously copyrighted maps or satellite images created using proprietary data, such as Google software (Google Maps, Street View, and Earth). For more information, see our copyright guidelines: http://journals.plos.org/plosone/s/licenses-and-copyright.

(1) You may seek permission from the original copyright holder of Figure 1 to publish the content specifically under the CC BY 4.0 license.  

**Additional Editor Comments:**

Your study is well written. Please carefully go through the peer review reports and make those changes. This will greatly enhance the readability of your paper. 

Reviewers' comments:

Reviewer's Responses to Questions

**Comments to the Author**

1. Is the manuscript technically sound, and do the data support the conclusions?

Reviewer #1: Yes

Reviewer #2: Yes

2. Has the statistical analysis been performed appropriately and rigorously? 

Reviewer #1: Yes

Reviewer #2: Yes

3. Have the authors made all data underlying the findings in their manuscript fully available?

Reviewer #1: Yes

Reviewer #2: Yes

4. Is the manuscript presented in an intelligible fashion and written in standard English?

Reviewer #1: Yes

Reviewer #2: Yes

5. Review Comments to the Author

Reviewer #1: Thank you for the opportunity to review the article titled "Face mask use in the City of Chennai, India: Results from three serial cross-sectional surveys, 2021". The authors have tried to assess face mask usage during the COVID-19 pandemic in Chennai. I believe that the manuscript could benefit from the following suggestions:

1. Kindly rewrite the abstract, as the methods mentioned in the abstract is confusing, and please stick to the abstract guidelines specific for Plos One

2. Line 73, which reads "We also documented mask compliance in the city's shopping malls for the third time point (round 5) alone" is not appropriately placed and is confusing. How many rounds were conducted? 3 or 5? Please be specific and modify the statement.

3. Why mention about rounds 1 and 2 from 2020 in this study? when the objective is to estimate compliance with appropriate mask usage at three different time points in 2021?

4. Why were these specific time periods chosen for sampling? the justification for the final round is provided, please provide the rationale and justification for conducting the survey at specific time periods of 2021

5. Why were 32 streets chosen? and how were the 32 chosen from 45,000? what sampling strategy was followed?

6. Pharmacies and religious places had "no entry restrictions?" and thus considered indoor settings.? not even in pharmacies? a personal would insist on mask usage?

7. "We excluded areas with security personnel who insisted on wearing a mask before entry.. Then why include Malls? which mandates mask usage before entry?

8. How was it possible to record the mask usage and compliance among motorcycle riders, who wear helmets?

9. How did the estimates calculate for the population inside the moving bus? how did the data collector collect this information? Were they stationed in the street or inside the bus?

10. Table 1: Round 3 is March 2020? or 2021?

11. When the objective is not to compare with the 2020 estimates, I feel there is no necessity to compare with round 1 and 2 estimates..

12. Please check the referencing style.. eg 2,10,13 are not in Vancouver

13. Please provide clearer images, preferably in TIFF format

14. Why did the study observe higher compliance during July 2021, while the peak was in March - April 2021?

Reviewer #2: I would like to express my gratitude for the opportunity to review the manuscript titled "Face mask use in the City of Chennai, India: Results from three serial cross-sectional surveys, 2021". The study highlights the importance of developing more effective communication strategies on appropriate mask use for older age groups and addressing compliance issues in crowded indoor settings to enhance public health measures during a pandemic. The following are some suggestions that I've considered for the manuscript:

1. The line number 71, which reads “We estimated the compliance with appropriate mask usage in the outdoor and indoor locations of slums and non-slums of Chennai, Tamil Nadu, India, at three different time points in 2021.We also documented mask compliance in the city's shopping malls for the third time point (round 5) alone.” Please modify the statements for clarification and justify the inclusion of shopping malls in the survey.

2. Also, line number 118 states, “We excluded areas with security personnel who insisted on wearing a mask before entry” please justify the inclusion of shopping malls in the survey as these areas were with strict enforcement for wearing masks.

3.As the objective was to estimate the compliance with appropriate mask usage at three different time points in 2021, justify the comparison of findings from the round 1 and 2 surveys of 2020 in the current study.

4.Line 109 mention, Mask" was defined as any cloth mask, medical mask, or N95 respirator worn over the face. We described appropriate mask use as wearing a mask covering the nose, mouth, and chin and inappropriate use as wearing the mask either below the nose or mouth. When the individual did not wear a mask, we considered it no mask. Cite reference for the same.

5.Line 130, which reads, “We included pedestrians, bicycle riders, motorcycle riders, autorickshaws, and bus passengers, excluding individuals traveling in a car.” How was it feasible/possible to record the mask usage and compliance of helmet-wearing motorcycle riders? as well as the bus passengers inside the bus?

6.Line 145, “We classified the mask compliance in each of the four settings into three categories: appropriate, inappropriate, and no mask, and computed the proportion and 95% confidence interval (CI). But the findings on ‘inappropriate use’ and ‘no mask use’ category were not mentioned.

7.Line 127 states, “Upon reaching the specific tagged location, the data collector would observe the individuals crossing from right to left direction, one individual at a time, and record the individual’s mask compliance. We maintained this directional rule to minimize possible duplication. Also mention the status of the observer as it may lead to duplication of the observation - whether stationary observer measured adherence by passersby, or a mobile observer measured adherence in stationary people.

8.For the south region, the compliance was low in round 1 and 2 whereas it was high in round 3-5, comparing with other regions of the city. Justify the findings.

9.Please check for the year in Table 1 for Round 3. Was it March 2020 or 2021?

10.Please check the reference numbers 2,10 and 13 for its styling with reference to the journal guidelines.

11. Mention about the generalizability of the study findings? Also mention the strengths and limitations.

6. PLOS authors have the option to publish the peer review history of their article (what does this mean?). If published, this will include your full peer review and any attached files.

Reviewer #1: No

Reviewer #2: No

---

## [Author Response · Author response to Decision Letter 0]

1 Jan 2024

Thank you, Reviewers, for all the comments. 

We have tried our best to include and answer all these comments. 

Response to Reviewers is added as a separate file also.

---

## [Editor Report · Decision Letter 1]

16 Jan 2024

Face mask use in the city of Chennai, India: Results from three serial cross-sectional surveys, 2021

PONE-D-23-17287R1

Dear Dr. Sharan,

We’re pleased to inform you that your manuscript has been judged scientifically suitable for publication and will be formally accepted for publication once it meets all outstanding technical requirements.

Kind regards,

Vijayaprasad Gopichandran

Academic Editor

PLOS ONE
---

## [Editor Report · Acceptance letter]

27 Mar 2024

PONE-D-23-17287R1 

PLOS ONE

Dear Dr. Sharan, 

I'm pleased to inform you that your manuscript has been deemed suitable for publication in PLOS ONE. Congratulations! Your manuscript is now being handed over to our production team.

Kind regards, 

on behalf of

Dr. Vijayaprasad Gopichandran 

Academic Editor

PLOS ONE